# Situated Affectivity, Enactivism, and the Weapons Effect

## Michelle Maiese

Department of Philosophy, Emmanuel College, Boston, MA 02115, USA; maiesemi@emmanuel.edu

**Abstract:** Existing research on the "weapons effect" indicates that simply seeing a weapon can prime aggressive thoughts and appraisals and increase aggressive behavior. But how and why does this happen? I begin by discussing prevailing explanations of the weapons effect and propose that these accounts tend to be over-intellectualistic insofar as they downplay or overlook the important role played by affectivity. In my view, insights from the fields of situated affectivity and enactivism help us to understand how cognitive and affective processes jointly contribute to the weapons effect. Insofar as the presence of weapons alters subject's bodily-affective orientation and thereby brings about *embodied mindshaping*, it changes the way they engage with and understand their surroundings. To understand the weapons effect, we will need to examine the constitutive interdependency of appraisal and affectivity and the way in which they jointly motivate action. My proposed account emphasizes the role of affectivity in affordance perception and the way in which subjects gauge the meaning of an object according to its action-possibilities.

**Keywords:** affordance; aggression; enactivism; mindshaping; situated affectivity; weapons effect



## 1. Introduction

Existing research on the "weapons effect" indicates that simply seeing a weapon can prime aggressive thoughts and appraisals and increase aggressive behavior. Since the initial studies were conducted in the late 1960s, these experimental results have been replicated many times, both in the laboratory and out in the field. Such results point to an important sense in which *the gun helps pull the trigger* [1]. But how and why does this happen? I begin by discussing prevailing accounts of how the weapons effect occurs and propose that these explanations tend to be over-intellectualistic insofar as they downplay or neglect the important role played by affectivity. This is puzzling given that both real-life and fictional situations involving weapons often have great affective salience. It is important to consider how the bodily-affective arousal associated with fear, agitation, or anger might help to focus a subject's attention, modulate their appraisals and interpretations, and motivate aggressive behavior. Can insights from the fields of situated affectivity and enactivism help us to understand how the thought and appraisal processes that contribute to the weapons effect are infused with affectivity?

Work on embedded and situated cognition and affectivity has explored the ways in which elements of the material world structure, scaffold, support, or distort these processes; work on enactivism has emphasized that cognition (sense-making) is fundamentally relational, action-oriented, and affective. I argue that an enactivist account of affectivity that views appraisal, thought, affectivity, and bodily arousal as constitutively interdependent is well-equipped to account for how the weapons effect occurs. My proposed account acknowledges that situational variables such as the presence of a gun do not simply influence someone's internal state, but also their mode of relating to the world. Insofar as the presence of weapons contributes to the affective atmosphere in which subjects are situated, it alters their bodily-affective orientation, brings about *embodied mindshaping*, and thereby changes the way they engage with and understand their surroundings. My proposed account also emphasizes the role of affectivity in affordance perception and the way in

which subjects gauge the meaning of an object according to its action-possibilities and what they can do with it, in light of their cares and concerns.

## 2. The Weapons Effect

In 1967, Berkowitz and LePage conducted an experiment to determine whether simply seeing a gun can increase aggression [2]. In their initial study, male college students were tested in pairs and evaluated each other's performance on a task (e.g., listing ideas a car salesperson might use to sell more cars). One of the participants was actually an accomplice of the experimenter who was pretending to be another participant. "Evaluations" of the task performance consisted of the number of stressful electrical shocks given, which ranged from 1 to 10. First, the accomplice evaluated the participant's performance by using either 7 shocks (provocation condition) or 0 shocks. Next, the participant evaluated the accomplice's performance. The participant was seated at a table on which was placed either (a) a shotgun and revolver, or (b) badminton rackets and shuttlecocks, and told that these items were part of another study and should be ignored. There was also a control condition in which no items were on the table. The study found that provoked participants who saw the guns behaved more aggressively (i.e., delivered more shocks) than the other participants.

Since then, researchers have found similar results in studies where pictures of guns were used, and in field experiments outside the lab. In a driving simulation experiment, participants were seated in a car that had either a handgun or tennis racket on the passenger seat. In the gun condition, experimenters told participants: "I told the other experimenter to clean up after himself, but he must have forgot. Please leave that gun exactly where it is. It is unloaded. It is for a different study involving police officers" [3] (p. 83). And in the no-gun condition, the experimenter told participants, "I told the other experimenter to clean up after himself, but he must have forgot. Please leave that tennis racket exactly where it is. It is for a different study". Presumably, this was done to instill a sense that the respective object just happened to be there, as opposed to being in the subject's possession or available for use. Participants drove significantly more aggressively when there was a gun on the passenger seat than when there was a tennis racket on the passenger seat [3]. For example, participants with a gun on the passenger seat were more likely to speed, tailgate, swear at other drivers, or use obscene language. In another study, Ariel and colleagues found significant increases in the use of force by, and assaults on, officers who were carrying TASERS [4]. That is, officers were significantly more likely to apply force when a TASER was present, and also significantly more likely to be assaulted by suspects when a TASER was present. It is not clear whether having tasers caused officers to be more confrontational or aggressive, or if the presence of TASERS caused suspects to behave more aggressively. However, the researchers note that "the available literature on police use of force points out that, on average, the causal mechanism [leading to aggression.] begins with the suspect rather than the officer" [4] (p. 296). This supports the hypothesis that merely seeing a weapon leads suspects to behave more aggressively. What is more, there is evidence that simply seeing weapons can impact people's beliefs about others. One study found that participants found a target person more disagreeable, hostile, and angry if they were holding items that could be used as weapons (e.g., garden shears) than other items, such as watering cans [5] (p. 34).

A 2018 meta-analysis [6] that integrated the results from all available weapons effect studies found that this effect was significant for both provoked and unprovoked participants, for both males and females, for participants of all ages, and even when toy weapons were used. This meta-analysis also sought to gauge the effects of weapons not just on aggressive behavior, but also aggressive thoughts, angry feelings, and hostile appraisals. It found that there was a significant effect of weapons on aggressive cognition, affect, appraisals, and behavior when the effects for these outcomes were combined [6] (p. 7). However, whereas the magnitude of the weapons effect was the largest for hostile appraisals, the presence of weapons did not significantly increase angry feelings.

Some theorists have suggested that the extent to which the weapons effect obtains has been over-estimated due to publication bias, i.e., the fact that studies which found little evidence of the weapons effect simply did not get published [6] (p. 13). During the 1970s, some studies [7] reported non-replication. What is more, while the (2018) meta-analysis examining the weapons effect [6] showed that the mere presence of weapons reliably increases the accessibility of aggressive cognition and hostile threat appraisals, the impact on aggressive behavioral outcomes was found to be inconclusive after taking into account publication bias. In addition, many past studies have been viewed as inconclusive and limited due to the small sample [3] and arguably small effect size [4]. Worries about the validity of weapons effect research is embedded in a context in which there is questioning about the replicability of classic research more generally [8].

Meanwhile, proponents point out that weapons effect studies have been replicated many times since 1967 and are in relatively good shape as far as empirical scientific methods and results go. It has been found to occur both inside and outside the lab, for many different kinds of weapons, and regardless of whether subjects were provoked [6]. In addition, all the average effect sizes from the 2018 meta-analysis were in the predicted direction, with weapons having a positive impact on aggression-related outcome variables. What is more, these effects are so robust that a tangible object need not be present to produce them; a visual cue of a gun, for instance in an image or video, is sufficient to generate the weapons effect. Such data suggest that people are sensitive to the appearance of weapons and become more likely to exhibit hostile perceptions and behavior. It may be that this effect does not hold in all circumstances, or that it's sometimes counterbalanced by other factors. Nonetheless, the fact that the presence of a weapon (whether an actual one or merely an image) often has a measurable impact on subjects' thoughts, appraisals, and behaviors warrants explanation. Indeed, given the real-world implications of these empirical data, we have reason to seek a deeper understanding of what is occurring so that we may begin to devise remedies or solutions.

## 3. The Weapons-as-Primes Hypothesis and the General Aggression Model

To explain how the weapons effect occurs, researchers in the 1990s proposed that it centrally involves a priming process [9,10]. According to the weapons-as-primes hypothesis, "the mere cognitive identification of a weapon increases the accessibility of aggression-related concepts in semantic memory" [11] (p. 309). "Weapon" concepts such as 'gun' and 'sword' become closely linked to aggression- and hostility-related concepts in semantic memory due to their similarity in meaning and their close association in experience. Concepts that frequently are activated simultaneously develop strong associations, so that the concept 'gun' becomes strongly linked to aggression-related concepts. Thus, once associations with the concept 'gun' have been formed, seeing a gun may increase the accessibility of associated aggressive thoughts and behavioral scripts. Increased accessibility of hostile or aggressive thoughts, in turn, may facilitate aggressive behavior, "by biasing one's interpretation of ongoing social interactions, or increasing the perceived appropriateness of an aggressive solution to a dispute" [11] (p. 308). According to this model, then, it is increased accessibility of hostile or aggressive *thoughts* that bias someone's interpretations and behavioral tendencies.

More recently, many theorists have looked to the General Aggression Model (GAM) to explain how the weapons effect occurs. This model is a framework for understanding why people behave in an aggressive manner and posits that two types of input variables can influence aggression. First, personal variables include individual characteristics such as gender, age, hormones, personality traits, attitudes, and values. Second, situational variables include a range of external factors that can influence aggression, such as exposure to aggressive cues, aversive events, and alcohol intoxication.

According to GAM, the variables that produce aggressive behavior do so by activating aggression-related cognitions, producing anger-related affect, and/or increasing arousal [12] (p. 50). Each of these three sorts of internal states can influence appraisal and

decision-making processes. Thus, this model treats aggressive thoughts, aggressive affect, and physiological arousal as three possible (and separable) *routes* to appraisal, decision, and action. Initially, there is an immediate appraisal of whether the situation is dangerous or threatening. Such an appraisal sometimes leads directly to automatic or impulsive behavior, whereas in other cases there is reappraisal of the situation (whereby the individual considers alternative interpretations and behavioral options). Subjects who form hostile appraisals, such that they interpret the actions of others as aggressive and expect them to respond in an aggressive manner, are more likely to behave aggressively. Thus, this model points to hostile appraisals (together with aggressive behavior) as the causal results of exposure to a weapon.

A 2018 meta-analysis [6] of the weapons effect focused on two of these potential routes to aggression: aggressive thoughts and angry feelings. It found that an insufficient number of studies examined the effect of weapons on physiological arousal, e.g., heart rate, blood pressure, and skin conductance. What is more, this meta-analysis revealed that the presence of weapons did not significantly increase angry feelings, leading Benjamin, Kepes, and Bushman to focus their attention on a cognitive route between exposure to weapons and aggression [6]. These theorists hypothesized that weapons would prime or activate aggressive thoughts and increase hostile appraisals. They also predicted that because provoked individuals become physiologically aroused and ready to attack others, there might be an even stronger weapons effect among provoked participants (so that they'd be especially prone to react angrily and aggressively). These predictions were borne out by the available studies. In addition, they observed that the magnitude of the weapons effect appears to be increasing over time.

In my view, existing discussions of both the weapons-as-primes hypothesis and the GAM do not adequately consider the role of affective experience in producing the weapons effect. Whereas these proposed explanations acknowledge that affective states can sometimes play a role in producing the weapons effect, they need not. What is more, researchers who have presented these accounts have focused on examining the role of aggression-related cognitions in producing such effects. They offer little in the way of explaining the linkages between these cognitions and a subject's affective orientation. First, the weapons-as-primes hypothesis points to associations between "weapon" concepts and aggression- and hostility-related concepts and rightly notes that these concepts become associated largely by way of experience. Indeed, such associations are likely to develop because many people's experiences with guns occur in situations where a gun is used to threaten or harm someone or from watching or reading stories in which gun violence occurs; and such experiences have an integral affective dimension. When subjects encounter guns that are being used to threaten or harm someone, whether in reality or in fictional accounts, this may very well arouse affective states of fear, agitation, hostility, or anger. These differing experiences are all likely to be highly affectively valenced and involve significant bodily arousal, such that the same kinds of bodily-affective feelings that subjects exhibit when someone is harmed also arise when a weapon concept such as 'gun' is activated. Subjects' affective condition, in turn, may very well influence their interpretation of social interactions, modulate their sense of what actions are appropriate, and bias them toward aggressive modes of response. An adequate explanation of the weapons effect needs to acknowledge the role of affective feelings and consider how such feelings are intertwined with subjects' appraisals and aggressive behavior.

It may seem that the GAM offers a fuller explanation insofar as it acknowledges that aggressive thoughts, angry feelings, and physiological arousal all can serve as possible routes to aggression. What is more, it acknowledges that these three possible routes to aggression "are not mutually exclusive or even independent" [6] (p. 3). Bartholow and colleagues rightly note that "the interaction among these aspects of the internal state influences appraisal and decision processes" [12] (p. 50). And Benjamin and Bushman point out that someone who has aggressive ideas might also feel angry and have elevated blood pressure [13]. However, these theorists have said very little about the nature of

the interdependency among aggressive thoughts, feelings, and bodily arousal. Also left unexplained is just how it is that "internal states" go on to influence appraisal and decision and whether these are simply one-way causal relationships.

What is more, the way that the weapons effect has been studied appears to reflect an intellectualist or cognitivist bias. Researchers working with the general framework of the GAM have focused on examining how weapons can prime or activate aggressive thoughts and increase hostile appraisals. They justify their research focus on the cognitive processes at play (rather than the affective processes) by pointing out that the few studies on affect that have been done seem to show that the effect of weapons on aggressive affect is not as strong as the effect of weapons on aggressive cognition [5]. If this is true, then perhaps the continued focus on aggressive thoughts (and the tendency to treat them as separable from angry feelings) in weapons effect research is reasonable and justified?

It is worth noting that the relatively few studies of the role of aggressive affect in the weapons effect have been based on self-reporting and measured using mood scales. In such studies, participants rate how they feel at that moment using a list of adjectives (e.g., ANGRY, FURIOUS, IRRITABLE) [14]. Meanwhile, researchers have measured aggressive thoughts using reaction times to aggressive or nonaggressive words or images [15]; and they have measured hostile appraisals by way of fist clenching [16] and by speed of identification of weapons v. neutral objects [17]. This focus on pre-reflective expressions of aggression or hostility does seem appropriate given that the weapons effect occurs without subjects being consciously aware of what is happening. However, whereas aggressive cognition and appraisal have been measured by way of pre-reflective indicators (which frequently operate below the level of participants' explicit awareness), aggressive affect has been measured by indicators at the self-reflective level. This is problematic given that there are some factors that make it unlikely that subjects will deliver accurate reports about their own affective condition.

For one thing, it is possible that participants simply lack awareness of their affective states, since such feelings are likely to operate below the threshold of self-reflective awareness. In addition, given social norms and perceived expectations, it is possible that subjects simply are unwilling to admit to researchers that they feel angry or irritable. Thus, the fact that participants do not rate themselves as angry or furious hardly shows that they are not experiencing those affective states or associated states of bodily arousal. Perhaps they experience occurrent anger, but do not identify it as such or report it accurately. What is even more likely, however, is that subjects experience a more diffuse mood and associated arousal; since their mood does not have any clearly defined intentional object or target, they are unlikely to report it. Whereas bodily expressions such as fist clenching have been treated as signs of hostile appraisal [16], they also might be plausibly viewed as signs of angry affect. In addition, it is possible that affective bodily feelings associated with anxiety, fear, or agitation (rather than anger) sometimes play an integral role in producing the weapons effect insofar as such feelings help to comprise subjects' immediate sense that a situation is dangerous or threatening. Alternatively, feelings of pleasure and excitement might contribute to someone's sense that a situation affords them power, domination, or control of others. However, there have not yet been studies to examine whether these different sorts of affective feelings play a role in generating the weapons effect.

My central worry, though, is that by supposing that participants can exhibit aggressive cognitions and appraisals in the absence of aggressive affect, these studies reinforce the notion that these components are separable and can operate independently (even if they often are causally related). While these models allow for the possibility that affectivity plays a role, their focus is on the cognitive route to aggression. What is more, they approach the weapons effect as if the impact that thoughts have on appraisal and behavior can be investigated without considering the role of affectivity and associated bodily arousal. The supposition that thoughts and appraisals are not deeply infused with affect, and that they can operate separately from bodily-affective feelings, obscures how the weapons effect

occurs; overlooked are the links between feelings, physiological arousal, appraisal, and aggressive behavior.

## 4. Situated and Enactive Affectivity

In this section, I discuss how research on situated affectivity and enactivism can help to shed light on how the weapons effect occurs and the way in which subjects' affective condition is linked to their thoughts, appraisals, and action tendencies. Here, I use the term 'cognition' in the way that researchers examining the weapons effect commonly use it, as referring to thought, appraisal, and judgment. Affectivity, as I understand it, encompasses the various ways in which an individual cares about objects, events, states of affairs, other people, their own life, etc. This includes occurrent emotions such as anger or fear, more diffuse mood states, concerns, and existential orientations [18]. An essential component of affectivity is bodily attunement, which anchors subjects in the world and makes the objects and situations they encounter intelligible insofar as they *matter* to them in some way or another.

The field now known as *situated affectivity* includes work on embodiment, enactivism, and extended affectivity and encompasses an array of views that emphasize the social and environmental embeddedness of affective experience. Some theorists interested in situated affectivity have adopted a user-resource model, according to which individuals use material and social resources in their surroundings to cultivate modulate, sustain, or enrich their experiences. This model aims to investigate how agents intentionally modulate their affectivity by deliberately constructing "affective scaffolds" [19,20] or using environmental structures and resources as "tools for feeling" [21] (p. 36). For example, people sometimes play particular songs or wear bright clothing to cultivate a happy mood. Specific material features of the environment with which subjects interact play a role in mood regulation and help subjects to establish particular sorts of organism-world relations [22]. (p. 1446).

But while the intentional use of environmental resources to shape one's own cognitive and affective processes is pervasive, this is not the only way in which aspects of the material and social world shape subjects' minds. As Slaby notes, an account of situated affectivity that focuses only on the deliberate use of resources neglects the way in which individuals' thoughts and feelings frequently are shaped by their surroundings without their even being consciously aware of it [23]. In fact, religious spaces, shopping malls, airports, restaurants, and office spaces often are designed to have such effects. Along these lines, Krueger and Osler discuss how some forms of social media clearly are designed to "engineer affect" and transform the affective mind-sets of their users [24]. However, in other instances, the modulation of affect can happen inadvertently, without being intended either by the individual being shaped or by the agents who designed particular social structures.

In addition, whereas some accounts have highlighted how material and social resources support, augment, and enhance affectivity and cognition, it is important to acknowledge that environmental resources also have the potential to distort these processes. According to Slaby's "mind-invasion-model," environmental structures and associated social practices sometimes "invade" subjects' affective lives in a way that harms them or goes against their interests [23]. Along similar lines, what Maiese and Hanna describe as "destructive and deforming mindshaping" molds subjects' cognitive and affective processes in such a way that detracts from human flourishing and makes it difficult for people to satisfy their needs [25]. Such mindshaping often occurs without individuals being aware of the nature or extent of this harmful influence. Acknowledging how affectivity is continuously modulated by the aspects of the material, interpersonal, and socio-cultural world should prompt a critical investigation of these environmental influences.

Building on these ideas, it is plausible to suppose that many material artifacts function as "affective affordances" [26] insofar they present occasions for getting affectively involved or prompt specific forms of affective engagement. Artifacts such as guns, for example directly modulate certain kinds of brain and bodily processes and thereby bring about embodied mindshaping [27], so that subjects' altered bodily-affective condition modulates

their sensemaking and worldly engagements. Indeed, there are instances in which guns are introduced deliberately, as a means to prompt certain kinds of affective experience. Hollywood, for example, recognizes the power of guns to enhance violent narratives [5] (p. 31) and shape the meaning of the events that unfold, and it's likely that they have such power in large part due to their affective salience and the fact that they are dangerous/pose a threat to safety and well-being. However, it is important to acknowledge that the presence of guns can impact subjects' affective condition inadvertently, without their being aware of it, and in ways that run counter to anyone's plans or intentions. As I discuss in the next section, the weapons effect provides a powerful example of such influence.

How is this account of embodied mindshaping different from the weapons as primes hypothesis? Whereas the weapons as primes explanatory model treats guns as stimuli that prime or causally trigger aggressive responses (in the form of thoughts, feelings, or arousal), my proposed account says that these different cognitive and affective elements are interrelated and embedded in a particular context. A context might be understood as the more general surroundings in which a particular situation unfolds, and which involves an overall mood [28] or affective atmosphere [29] (p. 78). Slaby, Mühlhoff, & Wüschner [30] use the term 'affective atmosphere' to highlight how salient elements of spatial settings and local arrangements help to generate a particular sort of mood or affective tonality. An affective atmosphere typically involves a range of contributory elements, including material, social, and discursive components. Here I focus on the potential role played by especially salient material artifacts, such as weapons Thus, rather than taking guns to be simply "aggressive cues that automatically and unconsciously elicit aggression" [5] (p. 30), as the weapons as primes hypothesis does, my proposed account says that they make a significant contribution to the overall affective context or atmosphere in which subjects are situated. In many cases, they help to create a general mood of agitation, unease, or hostility, which in turn modulates subjects' sense-making and agency. This may be especially true in the United States, where both guns and gun-related deaths are widespread.

In addition, a growing number of theorists maintain that affective states should be understood as enactive, i.e., as ways of engaging with and making sense of one's surroundings. The basic idea is that affectivity helps subjects to make sense of the objects and situations they encounter and to apprehend their surroundings as an arena of possible projects and goals. Thus, these feelings are constitutive of the sense of personal significance that objects, events, situations, other people, etc. involve. Insofar as affective states involve an element of appraisal, and thought and appraisal are fully bound up with bodily affectivity [31,32], the evocation of particular feelings of caring can "alter the informational character" of a subject's environment [33] (p. 470).

This account of affectivity draws from so-called 'autopoietic' or 'autonomic enactivism' [34,35], which aims to conceptualize how mindedness emerges in the natural world and emphasizes the biological character of mentality; as Thompson [35] puts it, mind is in life. Self-organizing, dynamic, living organisms enact meaning via continuous reciprocal interaction with their environments [35] (p. 79). Because a living organism always needs to maintain itself and supplement the autopoietic process with what it lacks to remain viable, stimuli acquire meaning to the extent that they relate positively or negatively to the "norm of the maintenance of the organism's integrity" [35] (p. 70). A norm of maintenance can be understood as an organism's optimal conditions of activity and its proper manner of realizing equilibrium within its environment. Adaptivity is a matter of being tolerant to changes by actively monitoring perturbations and compensating for them [35] (p. 147). Self-regulation and self-maintenance require that the living organism continuously exchange matter and energy with its environment, so that its patterns of worldly interaction are inherently bound up with its own viability constraints.

What some enactivists term 'sense-making' is the interactive process whereby living organisms interpret environmental stimuli in terms of their "vital significance." The constant regenerative activity of metabolism appears to endow living organisms with a minimal "concern" to preserve themselves and stay in existence, so that the environment

becomes a place of attraction or repulsion. This perspective changes the world from a neutral place to one that always means something in relation to the organism [34] (p. 188). What counts as a useful resource depends on the organism's structure, needs, and the way that it is coupled with its surroundings, so that "an organism's world is primarily a context of significance in relation to that organism's particular manner of realizing and preserving its precarious identity" [36] (p. 444).

Like Colombetti [37], I understand the sense-making of living organisms as simultaneously world-directed (intentional) and affective. Bodily feeling and responsivity are linked, at a basic level, to an organism's biological impulse to renew its own matter, regenerate, and go on living. Because the identity of an autonomous system is sustained under "precarious conditions" and the basic concern of a living animal is to survive, it develops a unique perspective on the world. While the discriminative capacity that allows a living organism to monitor and regulate itself with respect to its conditions of viability certainly is cognitive, it also is an affective-evaluative capacity that involves the living organism being "affected or struck by the suitability of an event for its own purposes" [37] (p. 19). What is conducive to survival and well-being is context-sensitive and agent-relative, and thus very much a matter of an organism's current predicament, concerns, embodied feelings, and emotions. This basic mode of affectivity therefore might be described as a bodily "feeling of existence" [35] (p. 229) that is valenced. Valence can be understood as "the primordial constitution of self-affection as a dynamic polarity, as manifesting itself in the form of a tension that takes several forms: like-dislike, attraction-rejection, pleasure-displeasure" [38] (p. 70). Bodily-affectivity and cognition are thereby connected at a more basic biological level—via apprehensions of vital significance and a living organism's bodily sensitivity to features relevant to its own survival and well-being.

Building upon some of Heidegger's ideas, I hold that it is our capacity for cognitive, affective, and practical intentionality, i.e., "care," which allows us to apprehend the world "as a significant whole, an arena of possible projects, goals, and purposes" [39] (p. 289). Along similar lines, and drawing from Husserl, Thompson [35] describes affect as the allure of consciousness, the pull that an object given to consciousness exercises on the ego. Allure motivates attention and implies a "dynamic gestalt or figure-ground structure" whereby some objects emerge into affective prominence, while other objects become unnoticeable [35] (p. 374). This sort of allure operates pre-reflectively, outside of explicit self-awareness, and serves to immediately target and focus someone's cognitive attention.

The notion of affective framing [31,32] helps to capture the idea that bodily affectivity is integral to processes of selective attention and appraisal insofar as it permeates our interpretations and patterns of engagement. One way to describe an affective frame is as an "affective mode of presentation" whereby "significant events or states of affairs [are] disclosed through diffuse, holistic bodily feelings" [40] (p. 437). These bodily-affective feelings help to determine someone's focus of attention and allow them to discriminate, filter, and select specific features and considerations in accordance with what they care about. In its most basic form, affective framing is a low-level mode of appraisal that has to do with "ecological significance to the organism" and involves schematic evaluation that is spontaneous and below the threshold of awareness [41] (p. 89). This sort of pre-reflective appraisal is physically grounded in "organismic processes of self-regulation aimed at sustaining and enhancing adaptive autonomy in the face of perturbing environmental events" [42] (p. 27).

However, the development of more sophisticated modes of movement and bodily sensitivity corresponds to the development of more sophisticated modes of sense-making and affective engagement. Affective framing selectively attunes an animal to its environment and allows it to gauge which factors are relevant given its specific needs, bodily structure, ways of moving, and current situational factors [43] (p. 265). What things in the environment mean for a human animal has much to do with their needs, desires, and feelings of what matters. Even among very young humans, there is a link between emotional response, movement, and the capacity to distinguish between different kinds of

sensory stimuli [44]. Fear provokes avoidance or flight from particular objects or situations; joy and love induce bonding and closeness; and anger establishes, reinforces, or expands boundaries with respect to selected objects.

This enactivist account of affective framing can assist us in making sense of the idea that someone's affective orientation not only (i) modulates their appraisals and behavior insofar as it acts as an interface between a subject and their surroundings, "mediating between constantly changing situations and the individual's behavioral responses" [45] (p. 556); but also (ii) is inseparable from the appraisal process. What I am suggesting is that affectivity and appraisal are not merely causally related (as the weapons as primes hypothesis and GAM suppose), but *constitutively interdependent* [46]. The enactivist approach emphasizes that caring is both bodily and evaluative, and linked at a basic biological level to the impulse to stay alive and fare well. The cognitive-bodily interpretations that constitute affective framing help alert subjects to what matters by allowing them to be sensitive, in a bodily way, to various aspects of their surroundings. Thus, as Colombetti notes, it is a mistake to suppose that appraisal and bodily affectivity are "merely instrumentally related" [46] (p. 536).

Instead, to say that affectivity and appraisal are inseparable and constitutively interdependent is to suppose that the subject's appraising their surroundings in a particular way is partially constituted by their undergoing various bodily changes and associated feelings (e.g., alterations in heart rate, blood pressure, breathing, skin temperature, and the orientation and positioning of body parts); associated bodily-affective feelings, in turn, are partly a matter of their appraising their surroundings in that particular way. A subject's affective sensitivity to the dangers of driving in the snow, for example, has both a somatic aspect and also evaluative intentional content [47] (p. 59); and these two aspects are inseparable and interdependent. The bodily sensitivity consists of various changes in heart rate, blood pressure, hormones, skin temperature, and the orientation and positioning of body parts; and the appraisal dimension consists of the apprehension of relevance and significance. Thus, their evaluation of existing driving conditions, and the way that they "perceive" them, is comprised partly of various bodily-affective feelings and associated physiological dynamics. But rather than being the object of conscious awareness, a particular bodily condition is lived through in the very process of evaluating one's environment, so that affective states count as a bodily sensitivity to what is significant [46] (p. 543). Indeed, these bodily-affective feelings are that through which a subject makes sense of objects, states of affairs, and situations out in the world [18] (p. 44). Such considerations also suggest that affectivity is not simply an internal state, but a way of relating to, actively engaging with, and being situated in one's surroundings.

The notion that attention and appraisal have an integral affective dimension [44] (p.188) is supported empirically by the fact that the systems for appraisal overlap a great deal with the systems for arousal. Pessoa explores how cognitive and affective processing is integrated into the brain and claims that the cognitive and emotional contributions to executive control cannot be separated and that they conjointly and equally contribute to the control of thought and behavior [48]. Likewise, Lewis discusses how the sub-personal processes that underlie appraisal and emotion are a distributed network of self-organizing and mutually influencing brain and bodily processes [49]. Together with the amygdala, bodily arousal and endocrine activity help to maintain an organism's homeostatic equilibrium, enhance attention, and prepare the individual for action. The tight link between cognition, affectivity, and action also is supported by the fact that among infants, "attention reactions (the immediate focusing of attention on newly appearing stimuli) involve emotions such as interest, fear, and surprise [44] (p. 188). It appears that attention cannot be understood adequately without considering its affective dimension. Lastly, the work of Barrett and Bar suggests that bodily feelings and sensations signaling an object's salience or relevance assist in perception and object recognition from the very moment that visual stimulation begins [50]. As I will discuss further in the next section, such findings support the claim

that affectivity contributes to perception and shapes the meaning and significance of objects such as guns.

The claim that bodily affectivity and arousal are bound up with appraisal also is supported by other psychological experiments. In a study conducted by Dutton and Aron, a young woman asked male passers-by to fill out a short questionnaire. At the end, she wrote down her telephone number and told the participants to call her if they wanted to learn more about the experiment [51]. Whereas some of the men had been approached on a seventy-foot high suspension bridge that was narrow and swaying, the control group had been approached on a wide, solid bridge that was only three feet high. While only ten percent of participants from the control group called back, half of the subjects who had been approached on the suspension bridge called back [51] (p. 516). It appears that bodily-affective arousal modulated their appraisal of the young woman and their assessment of her attractiveness. Such research further supports the notion that mindshaping is fully embodied [27], and that individuals' cognitive and affective processes often are molded in an immediate way, directly through their bodies.

My proposed account also rightly emphasizes that affectivity and appraisal are inseparable, mutually modulating elements that jointly motivate action. Insofar as affectivity helps subjects to gauge what sorts of actions are called for, there is good reason to think that affectivity is integral to affordance perception. The notion of affordance is a theoretical concept introduced by J.J. Gibson that emphasizes the complementarity of the animal and the environment and the link between perception and action [52]. What the environment affords are "what it *offers* the animal, what it *provides* or *furnishes*, either for good or ill" [52] (p. 237). These action-possibilities are specified relationally, both in reference to the objective properties of things in the environment as well as the bodily structure and capacities of a particular animal. The world that an agent inhabits is "disclosed as a matrix of differentially salient affordances with their own structure or configuration"; a particular animal encounters this broad ensemble of affordances and evaluates them, "often implicitly and automatically," for relevance [53] (p. 4). Because they have different body features and capabilities, different animals will perceive different affordances while detecting the same information.

To make sense of this, Rietveld and Kiverstein distinguish between a landscape of affordances and a field of affordances [54]. While the *landscape* (or "total ensemble") of affordances is comprised of the entire set of affordances that are available to a particular agent in a given environment at a specific time, the *field* of affordances consists of the relevant possibilities for action that a particular individual is responsive to in a concrete situation. Someone's perception of what a particular object affords partly depends not just on their bodily structure, capacities, and needs, but also their interests, habits [55], and past experience [56] (p. 213). Taken together, these factors modulate someone's sense of what actions an object affords and how it can be used. And as some enactivist theorists have emphasized, this process of gauging relevance and perceiving affordances also depends significantly on affectivity. Rietveld, for example, rightly notes that "this process of being responsive to relevant affordances is inseparable from the individual's concernfulness" [56] (p. 219). Among the key factors that contribute to the meaning and relevance of affordances are an agent's cares, needs, and desires [54]. Insofar as a subject's enduring concerns and overall perspective help to determine which action-possibilities show up as relevant for a particular agent in their specific situation [56] (p. 219), gauging what an object affords goes beyond sensory perception.

## 5. Toward an Alternative Account of the Weapons Effect

I have suggested that prevailing accounts of the processes whereby the mere presence of weapons can influence someone's aggressive thoughts, appraisals, and behaviors say little about the role of bodily affectivity. Although theorists acknowledge that priming plays a critical role, they distinguish between thought-priming and affect-priming and focus their attention on the former. In my view, an account that focuses solely on concepts, associative

links, and the priming of aggressive thoughts overlooks the role of bodily affectivity. How, then, can insights from situated affectivity and enactivism help to shed light on the process whereby the weapons effect occurs?

Recall that according to my proposed account, there is a link between affectivity and selective attention. Some of the existing research on weapons appears to illustrate and support this claim. For example, there is evidence that individuals respond as rapidly to guns and knives as they do to venomous snakes and spiders. In one study, experimenters measured reaction times to threatening stimuli (guns and snakes), pleasant stimuli (food and money), and neutral stimuli (trees and couches). Participants responded fastest to the threatening stimuli, and as quickly to guns as they did to snakes [57]. In another experiment, participants were presented with an array of objects and instructed to search for guns, staplers, or knives. They were told to press a key if they detected one of the objects and another key if they did not. Participants located the weapons more rapidly than they located the nonthreatening objects [17].

Such results suggest that guns and other weapons often capture people's attention. This is consistent with the hypothesis that affective salience is closely bound up with selective attention, and that what people notice has much to do with their cares, concerns, and interests. Subjects are likely to care about guns insofar as they are deadly and afford threatening or violent behavior; that is, these weapons are highly affectively salient due to their "vital significance." Indeed, because subjects have a basic concern for their own survival and well-being, and weapons are dangerous, seeing a gun may contribute to feelings of fear or agitation. This may be connected to the fact that people's affective experience of guns is shaped by what psychologists commonly call a "fight or flight response." Indeed, the fact that guns capture attention to the same extent that dangerous animals do suggests that these weapons are framed as a potential threat or a source of harm. Now, subjects need not be self-reflectively aware that they are feeling hostile, fearful, or agitated or ascribe any label to their affective condition. Instead, these bodily feelings often operate at a pre-reflective level, in-and-through the subject's heart rate, blood pressure, hormonal state, and overall somatic condition. But even if these states of bodily-affective arousal remain outside the subject's conscious awareness, they play an important role in targeting and focusing their attention. This helps to explain why participants in studies of the weapons effect more quickly home in on aggressive objects or aggressive words.

Seeing guns also qualifies as an instance of embodied mindshaping [27] insofar as the presence of guns molds subjects' bodily-affective condition and thereby modulates their cognitive processes. These material artifacts are not simply inputs from the environment, but instead contribute to the general affective atmosphere in which subjects form thoughts, appraisals, and judgments, and carry out actions, and in which moods and other bodily-affective states emerge. Upon seeing a gun, subjects may become agitated, anxious, or angry, and associated bodily arousal leads to a shift in their overall somatic condition. Their heart rate and blood pressure increase and their breathing speeds up, even if every-so-slightly. The brain rapidly releases cortisol, adrenaline (aka epinephrine), and other hormones like serotonin, dopamine, and norepinephrine. High levels of dopamine, in particular, have been found to be linked to tendencies toward aggression, competitiveness, and poor impulse control [58]. This shift in their bodily-affective condition makes them more prone to aggressive thoughts and orients them toward aggressive action-tendencies. Subjects become more likely to frame situations as threatening, or to judge others as hostile or dangerous. That is, their bodily-affective orientation modulates processes of emotional attribution and shapes their interpretation of other people's emotions. The presence of the weapon helps to create an affective atmosphere in which they interpret their surroundings from the standpoint of the fear-frame or the anger-frame, and this appraisal of people and situations is fully bound up with their bodily-affective condition. And of course, "whether a person actually behaves aggressively in a given situation depends a great deal on how the person interprets that situation [13] (p. 47). Interpreting others as dangerous or angry disposes them toward aggressive behavior.

Note that a weapon can make someone more aggressive even in cases where it is concealed but in close proximity. One study found that motorists with a concealed weapon in their car were more prone to drive aggressively (e.g., make obscene gestures or tailgate) than motorists who drove without weapons in their car [59]. If my proposed account is correct, having a gun in one's vicinity may contribute to an affective atmosphere that cultivates a mood of agitation, unease, or hostility. Associated bodily-affective feelings, in turn, shape how they attend to their surroundings and partially constitute their inter-pretations and appraisals of the driving behavior of others. For example, such feelings might contribute to their belief that other drivers are slowing down or cutting them off on purpose, to make things difficult for them. Whereas researchers adopting the GAM predict a "cognitive route between exposure to weapons and aggression" [6] (p. 350), my proposed account posits that affectivity necessarily plays a role, and is, in fact, inseparable from this "cognitive route".

My proposed account is fully consistent with the fact that when study participants become suspicious about being deceived (and more attentive to what is happening), or be-come aware of the hypothesis being tested, researchers do not observe the same aggression effects [6]. This is likely because the weapons effect is driven by affective framing processes that are operating at a pre-reflective level; when subjects become more explicitly aware of their situation, the affective atmosphere that the gun helps to create may begin to dissipate, so that the weapons effect is largely suppressed. In addition, research has shown that aggression effects tend to be larger in the lab, where conditions are more tightly controlled than they are in the field. This is to be expected given that a subject's overall affective condition is shaped by a number of different variables. My proposed account predicts that if past studies were modified so that some study participants saw a gun while listening to soothing music, the observed weapons effect would decrease or disappear, rendering these subjects less prone to aggressive behavior. This is because listening to soothing music might serve to counterbalance the heightened affective arousal they experience when they see a weapon and instead contribute to a calmer affective atmosphere. Indeed, understanding the weapons effect in terms of the generation of an affective atmosphere may help to explain why the weapons effect is not observed under all conditions. This points to a need for further empirical investigation of the various worldly elements that can contribute to a general atmosphere of hostility, fear, unease, agitation, and distrust, *or* help to dissipate it.

In addition, my proposed account, with its emphasis on affectivity, helps to make sense of why a difference in background knowledge and experience results in differential displays of the weapons effect. An important thesis in social and cognitive psychology is that "different people can perceive the same objective stimulus differently depending on the subjective meanings they attach to it" [12] (p. 49). These meanings often derive from someone's personal history and their particular knowledge about the relationships among objects in the world. Two relevant sorts of "knowledge structures" that Bartholow and colleagues highlight are (a) *perceptual schemata*, which include information about both nonsocial objects and social events, and (b) *behavioral scripts*, which contain information about how people ordinarily behave under varying circumstances [12]. Repeated exposure to the use of guns for aggressive purposes may lead people to form gun-related knowledge structures that include the idea that guns cause or enable aggressive behavior (perceptual schema), and also that guns often are used to threaten or harm people (behavioral script). The presence of a gun activates these gun-related knowledge structures, and "repeated activation of this link should increase the spontaneity with which evaluations or affect are associated with that knowledge" [12] (p. 49). What my proposed account adds to this story is that the way in which a subject is affectively situated depends not just on features of the environment, but also on their own individual orientation, background knowledge, and past experiences. That is, whether and to what extent weapons help to create an affective atmosphere of unease or agitation depends partly on the subject in question. As Anderson notes, atmospheres cannot be reduced either to human subjects or the material environment, but instead are constituted in dynamic unfolding relations between the

two [29]. Someone with a history of gun-related trauma might be especially susceptible to the affective atmosphere of agitation that the presence of a shotgun potentially helps to create, whereas the affective condition of a hunter who uses this sort of gun for sport may not be impacted in the same way or to the same extent.

The notion that the very same object might be "perceived" somewhat differently by different subjects also resonates with ecological psychology and the key notion of "affordances" discussed earlier in this paper. It also resonates with the enactivist notion that meaning is constituted dynamically via a living animal's ongoing, active, embodied engagement with the environment [60]. Recall that a field of affordances is the smaller subset of affordances offered by the environment that stand out as relevant for a particular agent in a specific situation. This field is defined in relation to a particular subject and can be understood as the "situation-specific, individual 'excerpt' of the general landscape of affordances" [61] (p. 7). Among the subjective factors that shape an agent's field of affordances are their body scheme, needs, and concerns. Which actions an object (e.g., a gun) affords will differ, then, depending on someone's capabilities and interests, as well their sense of the meaning and significance of that object. A subject's affordance field also will differ depending on how they are affectively situated in the world, insofar as this mode of situatedness partially comprises their appraisals of object, events, people, and situations.

Along these lines, Berkowitz proposed that weapons can have different meanings for different individuals and hypothesized that a weapon is apt to evoke aggressive actions to the extent that it has aggressive meaning [62]. Berkowitz writes: "Hunters might conceivably view guns as objects [that] they use only for sport (and not for hurting other people), so that they are reminded of the fun they have on autumn weekends when they hunt for wild game" [63] (p. 83). However, others who do not hunt for sport will be more likely to assign aggressive meaning to weapons and to have aggressive thoughts when they see a gun. Building on these ideas, Bartholow and colleagues sought to examine whether weapons priming effects differ for hunters and nonhunters [12]. They found that while pictures of hunting guns were more likely to prime aggressive thoughts among non-hunters, pictures of assault guns were more likely to prime aggressive thoughts among hunters. They also found that these differences in affective and cognitive responses to gun cues were linked to differences in aggressive behavior following gun primes.

Such results are fully consistent with, and support, my proposal that affectivity assumes center stage in both the "knowledge structures" that help to shape the meaning of guns and the process whereby guns prime aggressive thoughts and appraisals. These "knowledge structures" do not consist in a detached or abstract sense of what objects are, but rather a more practical sense of what one can do with those objects, i.e., what they afford. Thus, the extent to which aggressive thoughts or negative affect are primed by the presence of guns depends significantly on affordance perception and someone's sense of available action-possibilities. And I have argued that affordance perception, in turn, depends significantly on someone's affective orientation. A subject's cares, concerns and interests modulate their sense of what a gun affords, what it's "for," and what sort of meaning it has. These concerns and interests, together with their abilities and past experiences, help to shape their field of affordances and highlight specific action-possibilities as relevant and salient. When hunters see hunting rifles, it is less likely to spark feelings of fear or agitation, since for them, such weapons afford sport and fun activity with friends. Assault rifles, in contrast, will be perceived as having a distinctly different set of affordances, and due to their expansive background knowledge of guns, hunters may be especially attuned to their potential to inflict bodily harm. Meanwhile, action-possibilities associated with sport and fun (which are highly salient for a hunter) are likely to lack interest, relevance, or salience for a non-hunter. For them, all guns afford actions that cause bodily harm, pose a threat, or ensure one's own safety and protection. Perception of these affordances is bound up with bodily arousal associated with affective states such as agitation, fear, or anger, which in turn modulate subjects' thinking and appraisal.

My proposed account also is well-positioned to make sense of why the weapons effect appears to be increasing over time and predicts that subjects in the United States are especially likely to exhibit increased aggression when exposed to a weapon. In 2007, there were 90 guns for every 100 citizens in the United States [64], and in 2017, there were 120 guns for every 100 citizens [65]. What is more, the rate of firearm deaths per 100,000 people rose from 10.3 in 2007 to 12 in 2017, with 109 people dying per day in 2017 [66]. As guns become more widespread and gun-related deaths become more common, the presence of a gun becomes even more likely to contribute to a general affective context or atmosphere characterized by unease, distrust, fear, anger, or hostility. Those situated in this sort of affective atmosphere are then more likely to appraise other people and situations as dangerous or threatening, and to respond accordingly.

It is worth noting that my proposed account of the weapons effect resonates with some of the findings from situationist research in psychology. Such research shows that seemingly insignificant and marginal aspects of someone's particular situation can have a notable impact on their behavior. One study found that customers at a shopping mall who used a public pay phone and searched the change flap for forgotten coins were significantly more helpful when they had found a dime [67]. Another found that people were more willing to participate in a study when they had unexpectedly received a cookie [68]. One plausible explanation is that finding a dime or smelling freshly bake cookies uplifts someone's mood, even if ever-so-slightly. This uplifted mood, in turn, modulates their engagement with their surroundings and makes them more attuned to prosocial action possibilities. Thus, their altered affective state is fully bound up with their appraisals and action tendencies.

One rather puzzling finding from the 2018 meta-analysis was that the magnitude of the weapons effect was larger for images of weapons than for real weapons [6]. Since commonsense suggests that actual guns would be more likely to contribute to an atmosphere of agitation or unease, such data may seem to create difficulties for my proposed account. However, it is possible that when subjects are presented with images of guns, e.g., in slides [16] in a laboratory setting, they become more of a focal point of attention. A gun in the passenger seat of a car, in contrast, remains more in the periphery of awareness, yet nonetheless helps to shape the overall affective context or atmosphere. Further research should be done to determine whether some of the predictions associated with my proposed account prove to be accurate.

## 6. Conclusions

If the account I have outlined here is correct, then the ubiquitous presence of guns in the United States is heightening bodily-affective feelings of fear, anxiety, agitation, anger, and hostility. These feelings go on to shape subjects' appraisals of other people and situations and to inform their sense of which action-possibilities are relevant. It is likely that this has a detrimental impact on people's minds and behaviors and that the weapons effect, therefore, qualifies as an instance of "mind invasion" [23] and destructive and deforming mindshaping [25]. Clearly, this has important implications for policy, law, and public safety.

If merely seeing a gun (or even an image of a gun) contributes to an atmosphere or context that orients people's bodily-affective condition toward hostility and aggression, gun possession actually tends to make everyone (including gun owners) less safe. Indeed, the account I have outlined here suggests that carrying a weapon inadvertently and unintentionally has a detrimental mindshaping influence on subjects' affective and appraisal processes. They are more likely to become agitated, fearful, or angry, and also more likely to become attuned to aggressive action-possibilities, putting both them and those around them at increased risk. Such considerations suggest that as a society, we'd all be better off without guns in our midst; the laws and policies that we enact should reflect this. What is more, we should think more carefully about how to reshape our material environment so that it contributes to less hostile, more pro-social affective atmospheres.

**Funding:** This research received no external funding.

**Informed Consent Statement:** Not applicable.

**Data Availability Statement:** Not applicable.

**Conflicts of Interest:** The author declares no conflict of interest.

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
