# Peer review of "Situated Affectivity, Enactivism, and the Weapons Effect"

_philosophies, doi:10.3390/philosophies7050097_

Round 1
Author Response
Many, many thanks for this feedback; it has helped to me to make significant improvements to the paper. Page and line numbers below refer to the ‘clean’ version of the paper. The main changes I have made include:
- I have added details about past studies of the weapons effect.
- I have said more about how my proposed account differs from the weapons as primes hypothesis and the GAM. In particular, I emphasize that my proposed account treats affectivity and appraisal as constitutively interdependent. Thus, affectivity is necessarily involved in the cognitive process whereby the weapons effect occurs.
- I introduce the notion of affective context/atmosphere to explain the way in which affectivity is situated and how material artifacts such as guns contribute to the affective atmosphere in which subjects are situated. My hope is that this helps to flesh out how insights from situated affectivity can shed light on the weapons effect.
Responses to Reviewer #1:
- One worry was that I have conflated ‘cognitive’ and ‘deliberate.’ I agree that this was confusing things, and also detracting from my main points. I have deleted the sentence that refers to subjects deciding to behave aggressively. Like existing accounts, my account describes the process as automatic and non-deliberate. What my account adds is an emphasis on how the appraisal process is necessarily intertwined with affectivity.
- I agree that Sterelny’s conception of ‘cognition’ is not helpful in this context. I changed this, noting that I am using the term ‘cognition’ in the way that researchers on the weapons effect use this term, to refer to thought, appraisal, and judgment (lines 273-275).
- The reviewer notes that there is no evidence that an individual cannot perceive an object without engaging a cognitive process that invokes concepts and other information. I completely agree. I do not deny that a cognitive process is at play. However, I argue that the cognitive processes that produce the weapons effect (in particular, the appraisal process) and affectivity are constitutively interdependent. I make this more explicit on p. 9, beginning on line 441).
- The reviewer suggests that I jump back and forth between internal accounts and external/situated ones. I have introduced the notions of context & affective atmosphere to make it clearer that I favor an internal account, and also use these notions to help contrast my own proposed account with the GAM and weapons as primes hypothesis. I explain these concepts on p. 7 and also incorporate them into the discussion at various points in the paper.
- The reviewer recommends that I say more about the weapons effect and the existing studies. I am grateful for this feedback; the addition of some more details about the studies on p. 2 helps to support the claim that the weapons effect is a phenomenon worth investigating further.
- I added some material in the final paragraph of section 2, as suggested, to consider why some believe that evidence for the weapons effect is not very strong. I also explain why others think that the evidence is compelling, and why there are pragmatic reasons for investigating further (given the societal implications). This discussion begins on line 99.
- The reviewer notes that in section 3, I criticize the weapons as primes hypothesis and the GAM because they don’t adequately consider the role of affective experience. According to the reviewer, it is not clear that these explanations are leaving out of the picture. I agree that these models leave open the possibility that affect can play a role. However, may main concern is that affectivity need not play a role, according to these explanations (line 182). What is more, the cognitive route to aggression is treated as if it can occur apart from/separately from affectivity (beginning line 252). I deny that this is possible.
- I also note that subjects may not be experiencing a negative emotion; they may be experiencing something more like a mood (line 245). This would help to explain why they don’t report it, since this mood lacks any sort of clearly defined about. What is more, I note that other sorts of affective states (beyond anger and fear) could very well play a role (beginning line 252).
- The reviewer suggests that in section 4, my discussion of affective affordances and embodied mindshaping don’t really explain what is happening. I hope that my suggestion that the presence of weapons helps to cultivate a particular sort of affective atmosphere helps to tie together these different pieces of the explanation and shed light on how the weapons effect occurs by appealing to key ideas from situated affectivity. (p. 7)
- I deleted the sentence about there being “no sharp divide between affectivity and cognition.”
- I suggest that guns are likely to capture attention due to their “vital significance” and have reworded some of the material on p. 11 to try to clarify my point that subjects’ framings of guns (as dangerous) are bound up with bodily affectivity.
- I appeal to the notion of affective affordance to explain why hunters seem to be affected differently by hunting rifles v. other sorts of guns). Hopefully this helps to make sense of why hunters “perceive” certain sorts of guns differently (p. 13). What is more, I note that the tonality of an affective atmosphere depends both on the subject and the environment, and there are a range of elements that contribute to an affective atmosphere. This can help to explain why the weapons effect may not be observed in all cases.
Reviewer 2 Report
This paper analyses the phenomenon of the "weapons effect" — which suggests that simply seeing a weapon can increase aggression — through the lens of situated affectivity and enactivism. The paper first introduces the weapons effect and describes prevailing explanations for it, focusing on the weapons-as-primes hypothesis and the General Aggression Model. The text then goes on to introduce the reader to basic concepts of the framework of situated and enactive affectivity, highlighting aspects that seem especially relevant to the discussion of the weapons effect. Finally, the author(s) introduce an alternative account of the weapons effect, based on concepts from situated affectivity and enactivism. The authors suggest that the presence of weapons alters a subject's bodily-affective orientation, brings about embodied mind shaping and changes the way a subject engages with and gauges the meaning of their environment.
I think this text makes an intellectually stimulating read. It introduces an original idea that will be of interest to a diverse audience.
I would however suggest the following changes:
(1) There is considerable controversy in the literature regarding the existence and the magnitude of the weapons effect. Whilst the author(s) mention this in passing in the text, I believe it would be important to more clearly highlight critical/skeptical perspectives in the introduction and explicitly cite the relevant literature. This will provide a more nuanced introduction.
(2) I think the paper would benefit from a clearer and structured description of what exactly the advantages of the proposed situated affectivity / enactivist account of the weapons effect are. Does this new model fill explanatory gaps the other accounts can't fill? If so, which are they? Does the model lead to different predictions than the other accounts, which could be empirically tested? If so, which are those? Similarly, I would like to see a clearer critical assessment of the weaknesses of the new account. What aspects of the weapons effect can it *not* explain? What counter-evidence / criticism could be brought forward against it and how would the authors defend their model against such attacks from critics?
(3) Following from point 2, maybe the authors could provide a table outlining the main features of the two prevailing models of the weapons effect and contrast them with their proposed account? This table could include underlying assumptions, postulated mechanisms, level of explanation, support by empirical evidence, explanatory strengths and weaknesses.
(4) The claim that high levels of dopamine have been found to be linked to tendencies toward aggression has to be supported by an appropriate empirical reference.
(5) There are mistakes in some sentences that need to be corrected. Examples include line 38 ("but their also"), 204/205 ("given that are" "unlikely that subject will deliver"), 284 ("it plausible"), 454 ("the dangerous of"), 510 ("what people notice as much to do"), 545 ("angry or angry").
Author Response
Many, many thanks for this feedback; it has helped to me to make significant improvements to the paper. Page and line numbers below refer to the ‘clean’ version of the paper. The main changes I have made include:
- I have added details about past studies of the weapons effect.
- I have said more about how my proposed account differs from the weapons as primes hypothesis and the GAM. In particular, I emphasize that my proposed account treats affectivity and appraisal as constitutively interdependent. Thus, affectivity is necessarily involved in the cognitive process whereby the weapons effect occurs.
- I introduce the notion of affective context/atmosphere to explain the way in which affectivity is situated and how material artifacts such as guns contribute to the affective atmosphere in which subjects are situated. My hope is that this helps to flesh out how insights from situated affectivity can shed light on the weapons effect.
Responses to Reviewer #2:
- I have expanded my discussion of some of the worries about empirical research on the weapons effect and explained why I think evidence for the weapons effect is worth investigating further, despite these worries, due to its societal implications. I also have added citations of some relevant literature. This discussion begins on line 99.
- At different points in the paper, I have elaborated on how my proposed account differs from the weapons as primes hypothesis and the GAM. In particular, I have appealed to the notion of an affective context or atmosphere (p. 7) to explain the sense in which affectivity is situated. I also have said more about the way in which affectivity and appraisal are constitutively interdependent (and thus not merely causally related or separable, as the existing models suppose).
- I note some of the predictions of my proposed account, in particular, how it can account for why the weapons effect seems not to hold under some conditions (line 633) I note that “This points to a need for further empirical investigation of the various worldly elements that can contribute to a general atmosphere of hostility, fear, unease, agitation, and distrust, or help to dissipate it.”
- I note that my proposed account also is well-positioned to make sense of why the weapons effect appears to be increasing over time. (line 712)
- I also say more about why hunters seem to be affected differently by hunting rifles v. other sorts of guns). (p. 13)
- I also note that one of the things my proposed account cannot explain easily is why the weapons effect was larger for images of weapons. I speculate about why this might be beginning on line 736.
- I did not add a table outlining the main features of the two prevailing models, partly due to space constraints. I also was unsure about how to construct a table that would encapsulate the main points.
- I added an empirical reference to support the claim that high levels of dopamine have been found to be linked to aggressive tendencies.
- I have done my best to correct typos and mistakes in the text. Thanks for your help!
Round 2
Reviewer 1 Report
This version of the paper is in better shape. I still have two general concerns, but maybe they don’t need to be fully resolved.
(1) I still think that pursing what is, at least in part, an externalist (i.e., situated, environmental, or ecological) account and contrasting it with the weapons-as-primes hypothesis and the general aggression model—which are clearly internalist (i.e., traditional functionalist) accounts—creates some tensions. At best, the author and those committed to either the weapons-as-primes hypothesis or the GAM may be talking past each other. At worst, the author’s account fails to explain the phenomenon for which an explanation is needed. So, where we want account of how we get from perceiving a weapon to aggressive behavior, the author just, for instance, invokes the “affective atmosphere” and has it cause the “agitation unease or hostility” (p. 12). Perhaps this is an issue, however, that can be worked out as the account is further developed and others respond to it.
(2) Section 4 could be tighter and better organized. There are a lot of concepts and ideas being introduced there, and, given my concerns mentioned in (1), I think it would be better to focus on just the tools needed for the account that is described in section 5.
(3) Some typos:
Page 1, line 38: “also their” is written twice.
Page 9, line 433: missing a word—“it,” I think.
Page 9, line 466: typo (‘ans’)
Page 11, line 558: missing a word—“on a key”?
Reviewer 2 Report
My comments have been sufficiently taken into consideration.
There are still a few minor mistakes in some sentences (e.g., line 38 ("also their also their"), line 64 (".."), line 82 ("leading to aggression."), line 274 ("commonly us it")), but these should be caught by the copy editor.